# *“I Did Not Believe You Could Get Better”*—Reversal of Diabetes Risk Through Dietary Changes in Older Persons with Prediabetes in Region Stockholm

**DOI:** 10.3390/nu11112658

**Published:** 2019-11-04

**Authors:** Linda Timm, Meena Daivadanam, Anton Lager, Birger Forsberg, Claes-Göran Östenson, Helle Mölsted Alvesson

**Affiliations:** 1Department of Public Health Sciences, Karolinska Institute, SE-171 76 Stockholm, Sweden; meena.daivadanam@ikv.uu.se (M.D.); Anton.lager@ki.se (A.L.); birger.forsberg@ki.se (B.F.); helle.molsted-alvesson@ki.se (H.M.A.); 2Department of Food Studies, Nutrition and Dietetics, Uppsala University, 751 22 Uppsala, Sweden; 3International Maternal and Child Health division, Department of Women’s and Children’s Health, Uppsala University, 751 85 Uppsala, Sweden; 4Department of Molecular Medicine and Surgery, Karolinska Institutet, Karolinska University Hospital, 171 76 Solna, Sweden; claes-goran.ostenson@ki.se

**Keywords:** risk, prediabetes, reversibility, lifestyle modification, habits, T2D

## Abstract

Diabetes risk can be controlled and even reversed by making dietary changes. The aim of this study was to improve the understanding of how older persons with a high risk of developing Type 2 diabetes manage and relate to information about diabetes risk over a ten-year period. Fifteen qualitative interviews were conducted among participants from the Stockholm Diabetes Prevention Program (SDPP). The participants were asked to recall the health examinations conducted by the SDPP related to their prediabetes and to describe their experiences and potential changes related to diet and physical activity. Data were analyzed using qualitative content analysis. The main theme found was that T2D (type 2 diabetes) risk is not perceived as concrete enough to motivate lifestyle modifications, such as changing dietary patterns, without other external triggers. Diagnosis was recognized as a reason to modify diet, and social interactions were found to be important for managing behavior change. Diagnosis was also a contributing factor to lifestyle modification, while prognosis of risk was not associated with efforts to change habits. The results from this study suggest that the potential of reversing prediabetes needs to be highlighted and more clearly defined for older persons to serve as motivators for lifestyle modification.

## 1. Introduction

Risk communication in health is about balancing uncertainties about prognosis and diagnosis, and the information about an uncertain risk prognosis is difficult to communicate [1,2,3]. Risk is not a stable condition, it is a multidimensional and dynamic interplay between feelings and rational understanding and varies between experts and lay persons [4]. Health professionals often use scientific language to convey, for example, statistical risk based on population calculations that allow patients to interpret the message in various ways and can lead to both over- and underestimations of the level of risk for the individual [2,3,5]. This risk communication skill is increasingly relevant in prevention strategies for non-communicable diseases (NCDs) to communicate risk, raise awareness, and encourage protective behavior [6]. One example is to promote preventive actions to reduce the onset of type 2 diabetes (T2D) [6,7,8] and the risk of developing prediabetes, a stage where a person has elevated glucose values but not yet high enough to be considered diabetic. 

Strategies of communicating the risk of NCDs vary between target groups and settings globally. A key prevention component is the provision of recommendations for lifestyle modifications communicated through lifestyle counselling from health providers and health promotion messages through print, audio, and visual media. In some countries, health promotion programs are reported to be covered at the primary health care level [9], while in other countries, both patients and providers experience a need for more lifestyle modification education [10]. Several countries also offer “preventive check-ups” to prevent cardiovascular diseases [11,12], and national screening programs have been established to prevent NCDs [12]. In Sweden, prevention programs for cardiovascular diseases have been implemented [13,14,15] as well as research studies including diabetes and prediabetes screening [16,17].

Diabetes prevalence is rising globally, and early detection is especially relevant for prediabetes and T2D, which usually have an asymptomatic latent initial period [18]. Screening to identify prediabetes is one prevention strategy which creates opportunities for the reversibility of high glucose values [19,20,21]. However, a negative screening test can be falsely interpreted as a low risk of developing T2D and can also translate into a lack of interest in lifestyle modification in people with elevated risk but no definitive diagnosis [19,22]. Persons with prediabetes can reverse the risk of diabetes by controlling elevated glucose values through lifestyle modification to become normal [23,24], which makes the reversibility important to communicate. The communication of opportunities for reversibility combined with clinical evidence of improvements in health status has also been found to be useful in other areas, such as in motivating smoking cessation [25]. Lifestyle modification, in terms of dietary changes and increased physical activity, is a key component for reducing the risk of developing T2D, as obesity is a major risk factor among multiple risk factors [6,7]. Other common risk factors are aging and genetic disposition [18].

Habits change over the lifecycle [26] but, in general, are well practiced behaviors that have become automatic actions [27]. When habits are integrated into a person’s daily lives, they become a part of their lifestyle. Research show that dietary habits often change after retirement [28,29,30]. Although it can be assumed that older people have difficulty in changing established lifestyle patterns, there is also the potential to change the lifestyles of older people through nutritional education [31]. Also, lifestyle interventions due to diet and physical activity are even more effective for older-aged persons with diabetes [32]. According to Furst et al., food habits are influenced by ideals, personal factors, recourses, social environment, and food context [26]. Social support also influences eating patterns [33], may have a positive impact on exercise habits [34], and can reduce obesity [35], in addition to inducing a sense of higher self-control [36]. 

Communicating about the risks of having a disease or its diagnosis may cause anxiety, psychosomatic symptoms, and experiences of illness; furthermore, sensitive topics such as lifestyle related behaviors can be interpreted as a threat to personal integrity [3]. Thus, information about being at risk and risk awareness can potentially cause harm to individuals. According to Sachs (1995), the identification of risk factors can be “sickening” [37], which is in line with Smith-Morris (2010), who stated that a “sufferers’ identity” can be shaped [38]. However, Andersen and Whyte (2014) argued that risk information is positive when individuals are given an opportunity to reflect and act upon the information they are given [39]. 

The benefits of obtaining a diagnosis are well studied [40], yet literature on the experiences of living with a known high risk of developing chronic diseases is limited. Qualitative research methods are preferred to understand factors motivating lifestyle modification. There are qualitative studies highlighting a lack of knowledge and uncertainty about what it means to have prediabetes and to receive advice about lifestyle modification [41,42,43,44], yet the focus on reversibility makes this study unique. In addition, most studies have either been cross-sectional or intervention studies, while this study uses a longitudinal design to study aspects of habitual changes due to aging when living at risk of developing diabetes over a longer time period. No similar prospective/longitudinal studies on lifestyle changes with a focus on the reversibility of prediabetes have been conducted in Sweden or elsewhere. The aim of this study was to understand how older persons with a high risk of developing T2D manage and relate to information about diabetes risk over a ten-year period.

## 2. Materials and Methods

### 2.1. Setting

This study was conducted in the Stockholm region of Sweden. The prevalence of diagnosed diabetes in Sweden is around 5%, of which 85–90% of affected individuals have T2D [45]. The prevalence rate of prediabetes is estimated to be 6% for persons aged between 35 and 55 years of age in Stockholm [23]. According to the national guidelines for diabetes care and prevention, care is mainly provided at the primary healthcare level by a team of doctors and nurses [46]. Tests included for both T2D and prediabetes are HbA1c, blood pressure, and anthropometric measurements such as weight and height. Cholesterol levels are measured yearly for T2D patients and every second year for prediabetes patients [47]. T2D patients are invited to two annual primary care medical visits, while persons with prediabetes are invited to one annual visit [47]. The diabetes nurses are expected to provide T2D and prediabetes patients with lifestyle counselling for self-management [46,48]. Visits to nutritionists and podiatrists are arranged at the primary care level, and patients are further referred to ophthalmologists and endocrinologists as needed [48]. 

The study was approved by the Stockholm Ethical Review Board (ID 2016-353- 32). Before each interview, the participant was asked to read through the information letter and give both written and oral informed consent.

### 2.2. Study Design and Participant Selection

This qualitative study was nested in the Stockholm Diabetes Prevention Program (SDPP), a longitudinal study implemented in five municipalities in the region of Stockholm to increase knowledge and awareness of risk factors linked to T2D [49,50]. At the start of the SDPP study in 1992, the selected municipalities had populations ranging from 21 to 35,000 inhabitants, each from diverse socioeconomic groups with the majority of individuals born in Sweden [49]. The program was designed with three data collection time points: (1) baseline examinations in 1992–1998; (2) first follow-up in 2005–2006; and (3) second follow-up in 2016–2017. Approximately 8000 persons born in Sweden were included in the baseline sample, 6082 persons were included in the 10-year follow-up, and finally, 3459 persons were included in the 20-year follow-up. At all three times points, participants were tested for diabetes and prediabetes [23,49,51]. The data collected included a standardized Oral Glucose Tolerance Test (OGTT), anthropometric measurements, and a comprehensive questionnaire about lifestyle factors.

The qualitative study was conceptualized by the authors HMA, LT, AL, and C-GÖ, and the reporting followed the consolidated criteria for reporting qualitative research (COREQ) guidelines [52]. Purposive sampling was used to identify study participants who had a high risk of developing T2D between the first and second follow up examinations. A set of predefined criteria ensured the selection of participants with experience of living at risk of developing T2D for as long a time period as possible. All participants included in the study had a family history of diabetes (FHD), which was assumed to be a contributing factor to knowledge on diabetes and diabetes risk awareness. Only participants with either reversed or progressed prediabetes at the second follow up were included in order to have the opportunity to compare potential lifestyle changes between the participant groups. The inclusion criteria were as follows: (1) enrolment in the baseline study from 1994–1996; (2) informed about prediabetes status in the first follow-up period (2005–2006); (3) FHD; and (4) completion of the 20-year follow-up at least three months before the qualitative interview was conducted. The exclusion criteria were as follows: (1) stable glucose values at the first and second follow up points and (2) T2D diagnosis before the second follow up (see Figure 1).

Data collection was initiated in February 2016 when 136 out of 512 persons who had been informed of having a high risk of developing diabetes ten years earlier were examined in a second follow-up assessment. Seventy-two of these persons were eligible for inclusion according to our predefined inclusion criteria, and 38 persons were excluded based on the exclusion criteria. The remaining 33 eligible individuals had either progression of prediabetes from the previous follow up or had reversed their risk of developing diabetes (see Table 1). 

In this study, prediabetes reversal was defined through a Normal Oral Glucose Tolerance Test (OGTT) (fasting plasma glucose < 7.0 mmol/L and 2 h plasma glucose < 7.8 mmol/L) [23] in accordance with WHO recommendations [53]. The value at the 20-year follow up was compared with the value at the 10 year follow up for persons without a diagnosis of diabetes or treatment with any antidiabetic medication in between. The progression of prediabetes for the purpose of this study was defined based on OGTT [53] and HbA1c [54] values at the 20 year follow up as
Impaired Glucose Tolerance (IGT) on OGTT (fasting plasma glucose < 7.0 mmol/L and 2 h plasma glucose > 7.8 and < 11.1 mmol/L) AND HbA1c > 42 mmol/molImpaired Fasting Glucose (IFG) on OGTT [fasting plasma glucose > 6.1 and < 6.9 mmol/L and 2 h plasma glucose < 7.8 mmol/L, if measured] AND HbA1c > 42 mmol/molNewly diagnosed diabetes (fasting plasma glucose > 7.0 mmol/L or 2 h plasma glucose > 11.1 mmol/L) without prior diagnosis of T2D or treatment with any anti-diabetic medication in the intervening period.

The participants were divided into four groups by gender and reversibility or progression of prediabetes status. Ten participants were selected systematically and interchangeably from each group in the order that they appeared on each list. The selected persons were contacted by telephone and invited to participate in individual interviews. Before each interview, the participant was asked to read through the information letter and give both written and oral informed consent. In a second step, five more participants who had reversed their diabetes risk were contacted and asked to participate in interviews to further understand the reasons for and strategies used to reverse their diabetes risk and to achieve adequate information power [55] in terms of the participants’ understanding of reversibility. All invited persons but one consented to participate in the study (see Figure 1).

### 2.3. Data Collection and Analysis

Fifteen semi-structured interviews were conducted with participants. All interviews were conducted and recorded in Swedish by LT, a researcher with a background in Occupational Therapy and Global Health. The interview guide was mainly developed by HMA and LT, and it was informed by the concept of risk [1,56,57]. Questions addressed the experiences of living at risk of developing T2Ds and if/how lifestyle habits had changed over time. The guide was pilot tested for clarity, relevance, and pertinence to the aim of the study and thereafter finalized by the research team. A social vignette was also developed (Appendix A) [58] and used in the interviews to encourage the participants to recall and reflect on lifestyle changes. Analytical memos were made directly after each interview to document observations and reflections captured during the interview. The interviews were transcribed verbatim in the order they were conducted. In addition, contextual understanding of the SDPP data collection process was gained through observations done by LT at a healthcare center during participant examinations. Conversations with the nurses between the examinations of SDPP participants generated information about the screening procedure. A series of interviews was also held with a person who had worked for SDPP since 2002, which deepened the understanding of the implementation of SDPP and provided us access to SDPP questionnaires and information materials used in the program over time. Also, a nutritionist from a diabetes team was interviewed to learn about diabetes care at a lifestyle clinic in the Stockholm region.

After the initial open coding of the ten first interviews by hand, the transcripts were coded in the qualitative software program NVivo 10 using the principles of qualitative content analysis [59]. After coding and categorization, one common theme emerged. Perceptions of risk between persons with reversed diabetes risk and progression of prediabetes were compared separately to find potential differences in active lifestyle modifications. In a second step, five additional interviews were compared and contrasted with the initial material. The majority of the categories from the first analysis remained. However, in the categorization of the new codes, some of the codes did not match the previous categories. Therefore, new categories related to partners’ health were formed. This differed to the data from the first ten interviews; hence, the first ten interviews were recoded. The notes from observations were used to recall the interviews. Co-author discussions were held throughout all steps of the analysis process.

## 3. Results

Fifteen participants were at risk of developing type 2 diabetes (T2D) at the first follow up (2005–2006). At the second follow up (2016–2017), four of the participants (4, 5, 6, and 9) had progressed in their prediabetes status towards a T2D diagnosis. One of the participants from the group with elevated glucose values (8) interpreted the information on glucose levels as diabetic values and managed to reverse the risk to normal values. Ten participants had reversed their risk, and three participants (2, 10, and 13) had reversed their risk after receiving another diagnosis than T2D. Three participants (1, 7, 11) could have reversed their risk because of their partners’ diagnoses that indirectly led to changes in the diets of the participants. One participant with progressed prediabetes (9) had a partner with a diabetes diagnosis but did not report on specific dietary changes other than eating less due to ageing (see Figure 2). 

Three categories emerged from the main theme: “T2D risk is not perceived as concrete enough to motivate lifestyle modification without other external triggers”. The categories were (1) T2D risk is not urgent enough to act upon; (2) adaptations in everyday life as a part of aging; and (3) diagnosis as a motive for change (see Table 2). When the reversed group differed from the group with increased glucose values, the differences were specified. The quotes presented are translated from Swedish.

### 3.1. T2D Risk is Not Urgent Enough to Act Upon

Participants perceived the risk of developing T2D as intangible over time. The dimensions included a sense of T2D being diffuse, unreal, and invisible. All participants recognized that they had a risk of developing T2D due to genetic predisposition. However, this risk was described as static and therefore just *“a fact to live with”* and *“to take as it comes”*. It was not perceived as something urgent to act upon. To have a risk without experiencing symptoms added to the feeling of risk as being something fictive. In contrast, physical symptoms, such as impaired vision or lack of sensations in the feet and legs were recognized as signs of warning:“I think I would worry if I felt symptoms, this with bad blood circulation and sensations in the feet… Impaired vision. Or something like that, but I don’t have these warning signs”. (ID: 6)

Some of the participants described the risk as the absence of disease. An example of this is a participant who explained being at risk as follows:“At first, I thought: Yes, but oh what a relief that I didn’t have it [diabetes] then. Then it wasn’t really a risk anyway. Because you don’t know. You can go and carry on something you don’t know about”. (ID: 1)

The test results were received with a feeling of relief for not having the actual disease. The elevated glucose values, indicating prediabetes, were not interpreted as abnormal. Some participants described the invisibility of T2D as an uncertainty about what it actually means to have the disease, since you can have it without knowing it. Even though the risk was not a major concern, it was still talked about as something that exists in the body, as something you carry, a concern you cannot neglect because you know it is there. However, since prediabetes was not equated with T2D, and rather seen as *“Then it wasn’t really a risk”*. The interpretation was that this level of risk was an insufficient reason for behavior change. 

During general discussions on the concept of risk in the interviews, many participants mentioned the risk of cancer without any questions or prompts. They compared diabetes to cancer, and the risk of cancer was talked about as being much more serious and threatening than diabetes risk:“For me, it [the risk of diabetes] is not so bad. The fear is when they say that you have a tumor, malignant.” (ID: 2)

A cancer diagnosis was described as more concrete than a diabetes diagnosis, and complications of diabetes were more distant. Furthermore, the diagnosis of diabetes was not associated with death but with necessary lifestyle changes. 

When comparing the information reportedly given by SDPP staff to the participants and the understanding of the test results as explained by participants during the interviews, it appeared that many participants had understood the level of risk in a different way than the SDPP staff intended. There were examples of both over- and underestimating the risk (see Table 3). Difficulties were most apparent in the communication about reversed risk information. “Normal levels” could be understood as no change in status since the first follow-up visit ten years earlier and were expressed as uncertainty about the possibility of getting rid of being at risk. However, according to the SDPP protocol, the term “normal values” means that a person does not have prediabetes anymore, which corresponds to a positive improvement in the person’s glucose values:“I was a little unsure of what I was told … I interpreted the answer of elevated values as it could mean normal, because it wasn’t a diagnosis. And it made me unsure about the answer. So, I did not dare to interpret it as anything else than that I remained at risk as before, because I didn’t believe you could get better. That you remain where you are. So that’s my interpretation.” (ID: 10)

Confusion about what the numbers of glucose values from the test results meant was expressed by participants. Also, the justification of several similar tests was not clear. An illustration of the confusion about having several numbers to assess is shown the following participant with prediabetes progression, who explained that he had had a health check-up during which he received recommendations on lifestyle modification. He mentioned prediabetes:*“Because if you have four values or five test results* [different types of blood tests]. *If one of them says that you have prediabetes, then I do not have it in my world, because there are the four other values.” (ID: 5)*

In general, the participants seemed to lose interest in the information once they received information about not having T2D. In the example above, the participant seems to remember hearing about having prediabetes, but his own conclusion is that he does not have the diagnosis of prediabetes because there were test results that could be considered normal.

Even though it was expected that the participants would have difficulty recalling information received at baseline and at the first follow up ten years earlier, the participants also had difficulty remembering what they were told about the test results three months earlier. Both information given about the diabetes risk and recommendations on changes in lifestyle for prevention of diabetes were not easy to recall. At the last follow up ten years ago, all participants with prediabetes received a booklet with information about T2D and prevention advice, but only a few recognized the booklet when it was shown to them during the interview. Nonetheless, all of the participants appreciated receiving information about their risk of T2D and described it as an opportunity to prevent T2D in theory, yet the majority did not make lifestyle changes after being informed about having an elevated risk. Participants appreciated being well informed, but they often mentioned that the information and recommendations on lifestyle modification were not new to them and that they were already familiar with the information. Repetition of this general information was not considered to be useful, and advice about the benefits of physical activity and healthy food was judged to be “*common sense*”.

### 3.2. Adaptations in Everyday Life as a Part of Aging

Half of the participants in the reversed group had actively changed their diets for other reasons than the risk of T2D. Age was often mentioned as a reason to change lifestyle. A decreased level of physical activity due to aging was mentioned, while some persons both changed their diet and increased their level of physical activity to reduce weight when getting older. Diminished health of a partner was also reported as a reason to change lifestyle habits. When talking about eating habits, the participants who lived with partners talked about “their” eating habits as something they shared and had together. There were several examples of couples supporting each other in relation to health, and there was a willingness to modify one’s own habits to support the partner. One example is this woman, who had changed her diet because of her husband’s T2D diagnosis. She took his diagnosis as a mission and made sure they received the support they needed for diabetes management:… “We have changed diet. (…) Yes and then I’ve been to the hospital when my husband … and I demanded that we needed a nutritionist.” (ID: 7)

Participants reported eating habits changing over time and with ageing. Almost all of the participants were retired, which had changed the structure of their days. They reported eating regular meals, and most of them perceived their eating habits as being healthier than before. Eating fish and vegetables was often mentioned as a way of showing awareness in choosing healthy foods, and eating less than before was mentioned as something positive. It was also recognized that eating habits change in different environments. An example is the following quote from a participant who said that it was easy for her to keep a healthy lifestyle when she was at home alone or with her partner but that it was more demanding when she was out at social events:“But then when I’m away, I don’t know what sort of eating monster that (laughing) that I am... but I’m trying to stay away from it.” (ID: 7)

The majority of the participants expressed that they were willing to follow advice from medical experts. They expressed a general trust in healthcare and thought that they received the care they needed at the point at which they needed it. Overall, they were satisfied with the care they received for other conditions and therefore believed they would get help for diabetes if needed. The following quote illustrates trust in healthcare and that the decisions on lifestyle are left to the medical providers:-“If you received information now that you are at risk, how do you think you would…?”-Then I would go to the doctor and let him decide.-But if he tells you: You are at risk?-Then he can tell me what to do.” (ID: 12)

### 3.3. Diagnosis as a Motive for Change

Most participants reported diagnoses as a reason for making changes in lifestyle, for example, smoking cessation after having a heart attack or pneumonia or avoiding red meat after breast cancer operation and treatment:“Yes, but it is when I have been diagnosed that I’m grasping it. That’s when I kind of have to put in the effort.” (ID: 8)

A life with a T2D diagnosis was assumed by respondents to be strict and regulated. Medication was seen as the main difference between having diabetes and being at risk of developing the disease. Even if most participants did not think diabetes was very serious and it was assumed that they could live a good life with diabetes as long as they followed the advice, they still thought they would be more motivated to change habits if they got the diagnosis. Many of the participants thought the risk of developing T2D was not worth paying too much attention to. Instead, they thought they would be forced to make changes in lifestyle when they received the real diagnosis. Even though the participants often expressed having diabetes as being stricter and more regulated compared with diabetes risk, they also assumed that it would not be a big problem to live with the demands of T2D. 

One of the participants who had a husband with T2D did not think it was much of a struggle for him. She was therefore not motivated to try to prevent the disease, but was aware that she would have to make changes in her lifestyle if she were to receive the diagnosis:“No, because I know that if I develop diabetes, it is possible to live a good life anyway. And at the same time when you come to that situation, perhaps you can think of keeping sugar values down more. To walk or move and then kind of eat up your own sugar, or not consume wrong foods.” (ID: 9)

On the other hand, another participant with a husband who has diabetes compared prediabetes and diabetes like this:“Yes, prediabetes feels calmer, but diabetes is more regulated, more on track and with insulin and so, more control throughout the day. So, it’s more structured. More locked. So, you should avoid getting there.” (ID: 7)

One participant said that she had known about the risk for over twenty years but had never done anything to prevent diabetes. When the results from the second follow up showed values that she interpreted as having diabetes, she contacted a nutritionist and managed to modify her diet and reverse the glucose levels to normal in the glucose tolerance tests in only a few months. The trigger for her was the test results regarding her diabetic glucose levels, and her reaction was that the old habits needed to be changed:“Then I thought: Now I will change this totally. Now I will turn this and see if it helps. Then I started quite hard with a lot of vegetables and skipped all the grease. And after some time, I went for new tests, and they showed them to me, and there was nothing.” (ID: 8)

More than half of the participants who had reversed their diabetes risk to normal glucose values reported other diagnoses (own or in a partner) as being triggers to change habits (see Figure 2). In the reversed group lifestyle modifications were more apparent than in the group with progressed prediabetes. However, perceptions of risk were found to be similar in both groups (see Table 3).

## 4. Discussion

The awareness of T2D risk over time can create opportunities for reversibility. In this study population of older adults at risk, many participants misinterpreted the information about their risk status (see Table 3). This was especially prominent in the participant group with reversed T2D risk. These misunderstandings in communication add to the difficulties in the translation of scientific statistical knowledge into the clinical situation of the individual [5]. Research shows that communication between doctors and patients can create misinterpretations [60]. The patients are not always honest about treatment compliance and doctors do not always provide the patient with the full information to prevent and mitigate his or her worry [60]. A reason as to why participants did not understand the risk information from the follow up ten years earlier could be due to an attempt from the SDPP staff to not cause worry for the participants when they reported the test results. On the other hand, if the risk information is not communicated as being a serious condition, attempts to reverse the risk through lifestyle modification might not be triggered. Although research has shown that persons who receive the information of being at risk through screening do not automatically engage in modifying their lifestyles [21,22], it is important to improve the communication between health professionals and patients. Prediabetes status needs to be more clearly explained to patients, and the potential of reversibility has to be emphasized to create opportunities to prevent T2D. 

The participants perceived diabetes risk as abstract and diffuse in comparison with diagnoses that were experienced as more structured and concrete. The motivation to modify lifestyle was triggered by diagnosis, while prediabetes was not seen as a “real diagnosis” and not an urgent enough motive for lifestyle modification. Diagnosis serves the purpose of identifying signs and symptoms, classifying what sort of disease the person suffers from, and defining pathological conditions and their need for a specific treatment [40]. A diagnosis can be a way for an individual to explain their situation to others and can, to some extent, also shape a person’s identity [40]. This was shown by modifications of lifestyle done by the participants as a consequence of diagnosis and points to the need to talk about prediabetes as a clear diagnosis and not only as a stage where you have a high risk of developing diabetes. Interestingly, the participants expressed that they had experienced continuity in healthcare and have trust in medical doctors. This trust in experts make diagnosis serious, and therefore, a label such as getting a diagnosis might be a trigger for lifestyle change. Since the lack of a clearly defined diagnosis has been shown earlier to not facilitate lifestyle change [22], the label of a clear diagnosis could be a way to motivate lifestyle modification, which finds support in this study.

According to Slovic (2004), persons mainly respond to risk either rational or emotional factors [4], and this was seen among the participants who dealt with their test results in different ways. Some ignored high glucose values because they lacked experience of physical symptoms and did not feel illness, while others modified their behaviors and requested new measurements to verify a change. By being aware of invisible conditions such as diabetes risk, individuals can act upon their condition and take control over their health [39]. The medical technology procedures in terms of screening tests provide people with information about invisible conditions such as risk for developing T2D [19,39] and creates opportunities to talk about prevention strategies to avoid the disease. As our results suggest, these measurements make the risk more visible [39]. The level of risk as a measurement is like a “formative process” and can shape our understanding of health and disease [37]. According to Andersen and Whyte (2014), participants are “subject to the conditions” of risk as well as “subjects that act upon the conditions” [39]. As shown, both of these situations were found in this study. The individual has the power to make decisions that can shape the numbers of glucose values [61]. As one of the participants in this study clearly stated, she reacted to the glucose values from the second follow up examination, since the glucose level was interpreted as diabetic. After active changes in lifestyle, she was informed in a follow up examination that her glucose levels had reversed as a result of her active lifestyle changes. The numbers provided validation of the modification of behavior [61]. This means that she was a subject that acted upon her condition [39].

Even if not many, some participants acted on the fact of being at risk as it would have been a diabetes diagnosis and changed their habits totally. This goes in line with Smith-Morris’ idea mentioned in the introduction that awareness of risk can shape a “sufferer’s identity” [38], where the knowledge of being at risk prolongs the actual period of experienced illness [37,38]. To stress the risk as more severe and serious could therefore lead to overdiagnosis [62,63] and medicalization [63,64], where the condition of being at risk can be experienced as “sickening” [37].

The participants also recognized that they had a risk of developing T2D due to genetic predisposition and described that there was not much to do about this risk. When genetic risk is perceived as something you have to live with, you assume that nothing can be done or that anything you do will have no effect—both of which are untrue. There are studies which show that “modifiable factors relating to body weight, diet, and physical activity are more likely to impact on glycemic traits than genetic predisposition during a behavioral intervention” [65]. Also, even if the state of being at risk is perceived as static because of their family history of diabetes, the actual risk varies with the fluctuation in glucose values. Therefore, the reversibility of the risk of contracting diabetes should be emphasized by health services.

Several participants did not understand the information about their risk level in the way it was intended to from the providers perspective. Particularly, there was confusion about what the numbers of glucose values meant in relation to the level of risk. Health literacy and numeracy skills are important for achieving good health [66,67]. Although the participants in this study did not have language barriers in the communication with the SDPP staff, it was unclear to the participants what the results, given as numbers, actually meant. This goes in line with research showing that even people with good literacy levels can also have problems with understanding numerical data and calculations [68]. Patients can be given complex numerical data from providers in relation to the diagnosis and management of disease that are beyond their capacity to understand [67,69]. It could be that the test results are not explained well enough for people to understand what being at risk means. On the other hand, estimations of risk levels are subjective and vary depending on a person’s idea of what these risk levels mean. The multidimensionality of the concept of risk allows one person to interpret a level of risk as low, while another interprets the same risk level as high [4], since the interpretation of risk is subjective and risk perceptions are dependent on many contextual factors as well as factors linked to the individual. In addition, the information of risk may, according to the literature, be interpreted differently by the health provider and the patient [2,3], and this could also be the case for the communication between SDPP staff and the participants.

The participants clearly stated that the information on how to achieve a healthy lifestyle was not new to them and was seen as “common sense”. This suggests that something more than formal information is needed to create a change in health behavior. Knowledge on what should be done was not enough to create changes in lifestyle in this study. Anderson and Funnel (2005, 2010) argued for the value of empowerment of patients in diabetes treatment [70,71]. This also applies to the prevention of disease for persons at risk of T2D and prediabetes. This study showed that the knowledge of the reversibility of diabetes risk can be unclear and that trust in healthcare is a necessary but not sufficient condition for behavior change. The persons at risk of T2D need tools and support for self-management. For example, in our study, partners were found to influence dietary choices. This goes in line with other research showing that the social environment is one of the areas that influences food choices [26,72]. Lifestyle modification as treatment should therefore not only target patients but also partners. Family members should therefore be included in lifestyle counselling to achieve sustainable habitual changes.

In relation to the strengths of the study, we interviewed a homogenous study population [55], which, at the same time, is a limitation, as all participants were born in Sweden and are thus not representative of the total population in the Stockholm region. Therefore, the transferability of the study is limited to similar settings, such as suburbs in Sweden with a socioeconomically similar population as the areas where the SDPP study was implemented. Yet, the study could potentially be transferable to other parts of Europe with similar settings. Another strength is that although qualitative studies exist on perceptions of prediabetes [41,42,43,44], to the best of our knowledge, our study is unique in exploring the dimension of living at risk over time and with a focus on the potential of reversibility. A number of limitations also exist. We only had access to reported SDPP communication about risk but no access to actual phone conversations between doctors and participants when the results from the examinations were communicated. We can therefore not know how the risk information was communicated to the participants, which could affect the trustworthiness of the data. On the other hand, there are data on the experiences of lifestyle modification and perceptions of risk that are not dependent on the exact information received through phone calls and the main conclusions remain. Recall bias [73,74,75] also needs to be highlighted, since the participants were asked to describe changes in habits and lifestyle over a time period of ten years. This also applied when the participants recalled the health examinations conducted by SDPP. Furthermore, the information received about the participants’ risk status was appreciated, but social desirability bias towards answering positively in order to please the researcher cannot be excluded [76,77]. However, indirect questioning was used in the interviews, which should had mitigated this type of bias [78].

## 5. Conclusions

The multidimensionality of risk makes it important to understand how people relate to specific risks such as developing T2D over time. Diabetes risk was not perceived as a strong enough matter to act upon for the majority of older adults interviewed. Risk alone was seldom a reason for modifying lifestyle, and changes in habits were often caused by external triggers such as diagnoses. Also, a partner’s diagnosis was reported as having influenced changes in dietary habits and lifestyle choices. The risk of T2D was experienced as intangible and abstract in comparison with diagnosis, which was described as structured and concrete. The results from this study suggest that diabetes risk and prediabetes need to be more clearly defined, with more emphasis on tools and support for reversibility to serve as motivators for lifestyle modification in adults. The mismatch in communication between health professionals and participants shows a need to explore other methods of communicating risk meaningfully and effectively.

## Figures and Tables

**Figure 1 nutrients-11-02658-f001:**
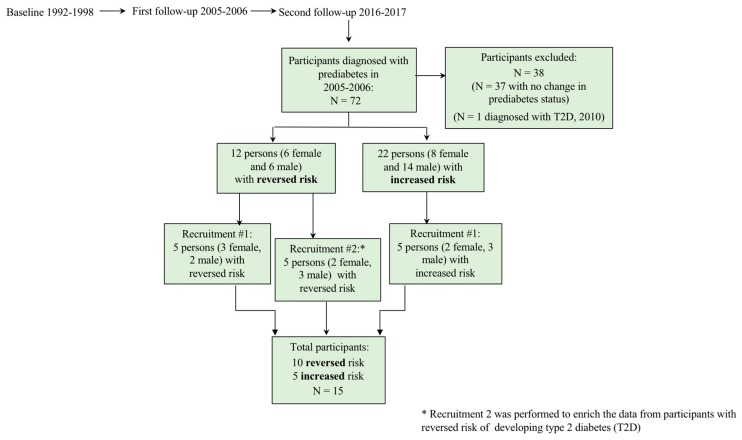
Selection of interview participants from the Stockholm diabetes prevention program (SDPP) register.

**Figure 2 nutrients-11-02658-f002:**
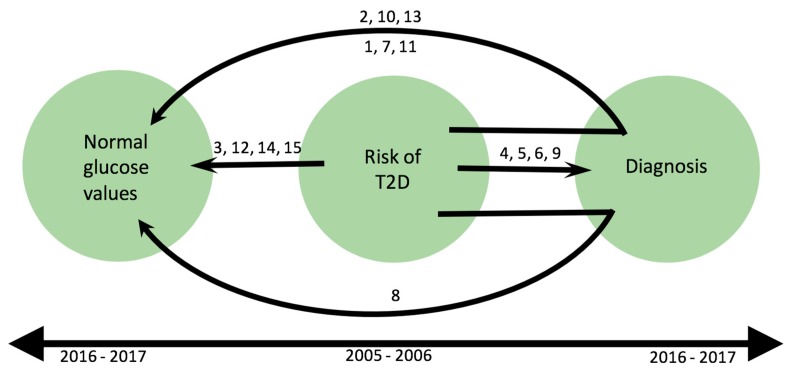
Graphical illustration of T2D risk in the 15 study participants over time. (Each number 1–15 represents a participant.

**Table 1 nutrients-11-02658-t001:** Characteristics of 15 interview participants with high glucose values (prediabetes) between years 2004–2005 and 2014–2017.

Participant ID	Sex	Age	Education Level	Household Status	Medical Status Reported by SDPP in 2014–2017
1	Female	68	Secondary/upper secondary school	Lives together with partner	Reversed T2D risk
2	Female	70	College/University	Lives alone	Reversed T2D risk
3	Male	72	Secondary/upper secondary school	Lives together with partner	Reversed T2D risk
4	Male	70	Secondary/upper secondary school	Lives alone	Progression of T2D risk
5	Male	71	College/University	Lives together with partner	Progression of T2D risk
6	Male	71	Secondary/upper secondary school	Lives together with partner	Progression of T2D risk
7	Female	67	College/University	Lives together with partner	Reversed T2D risk
8	Female	71	Secondary/upper secondary school	Lives together with partner	Progression of T2D risk
9	Female	68	Secondary/upper secondary school	Lives together with partner	Progression of T2D risk
10	Male	58	College/University	Lives together with partner	Reversed T2D risk
11	Male	70	Secondary/upper secondary school	Lives alone	Reversed T2D risk
12	Male	72	Secondary/upper secondary school	Lives together with partner	Reversed T2D risk
13	Female	71	Secondary/upper secondary school	Lives alone	Reversed T2D risk
14	Female	73	Secondary/upper secondary school	Lives together with partner	Reversed T2D risk
15	Male	72	College/University	Lives alone	Reversed T2D risk

T2D: type 2 diabetes.

**Table 2 nutrients-11-02658-t002:** Overview of the results process.

Theme	T2D Risk is not Perceived as Concrete Enough to Motivate Lifestyle Modification without Other External Triggers							
Category	T2D Risk is not Urgent Enough to Change Behavior			Adaptations in Everyday Life as a Part of Aging			Diagnosis as a Motive for Change	
Sub category	T2D is perceived as intangible	Difficult to understand what T2D risk means	Information is not enough to produce change	Common habits change together with others	Lifestyle modification trade offs	Responsibility of medical authority	T2D diagnosis as a reason to change habits	Other diagnosis than T2D as motive for change
Code	Risk is not concrete Prediabetes is not a real diagnosis Risk is invisible Absence of disease No fear of prediabetes	Uncertainty about what risk means Increased risk is normal Uncertainty about meaning of numbers Numbers as a tool for understanding risk information Risk is an abstract concept	Repetition is not useful Awareness of risk Obvious things Information is wanted but creates uncertainty Common sense	Eating as social activity Familiar preferences Shared responsibility Control for partner’s health Support each other	Weight reduction Aging Indulge oneself Pain Social events Willingness to replace diet Changes over time	Trust in healthcare Continuity in care Experts’ help Others positive experiences Believe in the doctors’ capacity Advice	Proof of having disease Medication Numbers make T2D visible Strict Fear of insulin injections Regulation	Disease is structured Diagnose Control Changes in diet is needed Changed level of physical activity because of diagnosis Partner’s diagnosis

**Table 3 nutrients-11-02658-t003:** Perceptions of risk and lifestyle modification over time.

Participant ID	Recruitment Status	Perception of Risk at the First Follow up 2004–2005	Lifestyle Modifications between First and Second Follow up	Perception of Risk at the Second Follow up 2014–2017
1	Reversed T2D risk	Yes, at risk	Active lifestyle changes because of T2D risk and weight reduction	Reversed T2D risk
2	Reversed T2D risk	Yes, at risk	Dietary changes because of stomach problems	Reversed T2D risk
3	Reversed T2D risk	No, not at risk *“It was ok”*	No active lifestyle changes	Do not know
4	Progression of T2D risk	No, not at risk *“I interpreted it as it was good”*	No active lifestyle changes	Increased T2D risk
5	Progression of T2D risk	No, not at risk *“No, I haven’t received a response”*	No active lifestyle changes because of T2D risk. Eats healthy and is physically active	No change
6	Progression of T2D risk	Yes, at risk	No active lifestyle changes	Increased T2D risk, but *“Nothing alarming”*
7	Reversed T2D risk	Yes, at risk	Active lifestyle changes because of T2D risk	Reversed T2D risk
8	Progression of T2D risk	Yes, at risk	No active lifestyle changes 10 years back, but active lifestyle changes after information from the last follow up	Increased T2D risk (interpreted as diabetic values)
9	Progression of T2D risk	Yes, at risk	No active lifestyle changes	Increased T2D risk
10	Reversed T2D risk	Yes, at risk	Dietary changes because of stomach problems	No change
11	Reversed T2D risk	No, not at risk *“Not at a particular risk”*	Dietary changes because partner’s change in choices of food	No change
12	Reversed T2D risk	No, not at risk	No active lifestyle changes	Reversed T2D risk
13	Reversed T2D risk	Yes, at risk	Dietary changes because of stomach problems	Reversed T2D risk
14	Reversed T2D risk	No, not at risk *“But maybe I have always had a little risk”*	Dietary changes to reduce weight	Increased T2D risk
15	Reversed T2D risk	Yes, at risk	Active lifestyle changes because of T2D risk	Reversed T2D risk

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
