# Peer review of "“I Did Not Believe You Could Get Better”—Reversal of Diabetes Risk Through Dietary Changes in Older Persons with Prediabetes in Region Stockholm"

_nutrients, 2019, doi:10.3390/nu11112658_

Round 1

Reviewer 1 Report

The paper is a descriptive report about the experience of 15 participants. The aim of the study is not clear. What do the authors mean "to improve understanding of how the older persons..."? Are the authors sure that the patients recall correctly the health examinations? What do the authors mean as "T2D risk is not concrete enough?" The authors can not conclude "reversing diabetes" because it is not possible to reverse diabetes, but only prediabetes. Eventually it is possible to reverse the progression of diabetes. A scientific analysis is missing.

Reviewer 2 Report

Title: 2 “I did not believe you could get better” – Reversal of 3 Diabetes Risk Through Dietary Changes in Older 4 Persons with Prediabetes in Region Stockholm

This is a qualitative study aiming to assess if perception of   T2D risk influences reversal or worsening of T2D risk.

The study has merit in its longitudinal design. Yet, it is limited by its small sample size. This reviewer, however, believes the study addresses with clarity how perception may lead to underestimation of risk in this population. Below are some comments, that if addressed may improve the manuscript. Thank you for the opportunity to revise this manuscript.

S 21:  please change dietary changes to include lifestyle modifications (i.e., diet and physical activity).

Introduction:  the introduction is clear.

Sentence (S) 97:  I would suggest the authors to change  from “ will explore” to explored.

S 95-96:  From the introduction, it is not clear how your study differs from existing literature. Are these other studies cross-sectional in design or while longitudinal, not as long as yours?  The authors need to strengthen the argument why their study is needed. In other words, make a better case. Also, the words “explore aspects” say very little. Again, expand more.

S145: needs to be reworded.

S149:  How much higher or lower glucose values needed to be in order to qualify as reversed or increased? S125 says OGTT’s were done. How OGTT data were used in exclusion and inclusion criteria? Did participants with impaired glucose tolerance or impaired fasting glucose were included. Please expand.

Round 2

Reviewer 1 Report

The authors have reviewed the work a adequately